# Determinants of skilled birth attendant delivery among reproductive age women in Lesotho: A multilevel analysis of demographic and health survey data

Hiwot Tezera Endale[1]*, Tseganesh Asefa[2], Tiget Ayelgn Mengstie[1], Mihret Getnet[3,4], Helen Lamesgin Endalew[5], Amare Belete Getahun[6], Gashaw Dessie[1], Thomas Kidanemariam Yewodiaw[7], Engidaw Fentahun Enyew[8,9]

**1** Department of Medical Biochemistry, School of Medicine, College of Medicine and Health Sciences, University of Gondar, Gondar, Ethiopia, **2** Armauer Hansen Research Institute, Addis Ababa, Ethiopia, **3** Department of Human Physiology, School of Medicine, College of Medicine and Health Sciences, University of Gondar, Gondar, Ethiopia, **4** Department of Epidemiology and Biostatistics, Institute of Public Health, College of Medicine and Health Sciences, University of Gondar, Gondar, Ethiopia, **5** Department of Surgical Nursing, School of Nursing, College of Medicine and Health Sciences, University of Gondar, Gondar, Ethiopia, **6** Department of Anesthesia, School of Medicine, College of Medicine and Health Sciences, University of Gondar, Gondar, Ethiopia, **7** Medical Officer at International Medical Corps, Amhara Region Emergency Operation Center, Gondar Field Office, Gondar, Ethiopia, **8** Department of Human Anatomy, School of Medicine, College of Medicine and Health Sciences, University of Gondar, Gondar, Ethiopia, **9** Department of Reproductive Health, Institute of Public Health, College of Medicine and Health Science, University of Gondar, Gondar, Ethiopia

* hiwottezera1@gmail.com

## Abstract

### Background

Skilled birth attendant (SBA) delivery, defined as childbirth assisted by trained health-care professionals such as doctors, midwives, and nurses, is a cornerstone of the World Health Organization's Safe Motherhood Initiative to reduce maternal mortality. Despite high SBA coverage in Lesotho, maternal mortality remains elevated. This study aimed to assess the determinants of SBA delivery among reproductive-age women in Lesotho using the 2023–2024 Lesotho Demographic and Health Survey data.

### Methods

A cross-sectional study design was applied using nationally representative data of 1,407 women aged 15–49 years. Multilevel logistic regression models were employed. Adjusted odds ratios (AOR) and 95% confidence intervals (CI) were reported, with p-values <0.05 considered statistically significant.

### Results

Skilled birth attendant (SBA) coverage was 91.8%. Women aged 35–49 had 60% lower odds of SBA delivery (AOR = 0.40; 95% CI: 0.17, 0.95). Higher maternal

**Data availability statement:** All relevant data are within the manuscript and its Supporting information files.

**Funding:** The author(s) received no specific funding for this work.

**Competing interests:** The authors have declared that no competing interests exist.

**Abbreviations:** ANC, Ante Natal Care; DHS, Demographic and Health Survey; LSDHS, Lesotho Demographic and Health Survey; MMR, Maternal Mortality Ratio; SBA, Skilled Birth Attendant.

education (primary: AOR = 1.47; 95% CI: 1.02, 4.90; secondary: AOR = 1.63; 95% CI: 1.12, 2.94; and tertiary: AOR = 3.88; 95% CI: 2.50, 5.20), wealthier households (AOR = 4.30; 95% CI: 1.10, 16.80), and age at first birth between 20–24 years (AOR = 3.74; 95% CI: 1.77, 7.89) were positively associated with SBA delivery. Women initiating ANC in the third trimester had 63% lower odds of SBA use (AOR = 0.37; 95% CI: 0.14, 0.99).

## Conclusion

Despite high SBA coverage in Lesotho, disparities exist, particularly among older women and late ANC initiators. Targeted policies such as community-based health education programs, transport support for rural women, mobile health reminders, incentives for facility deliveries, and the deployment of skilled birth attendants in underserved areas are vital to improving access, promoting education, encouraging early ANC engagement, and ultimately enhancing SBA utilization to reduce maternal mortality.

## Introduction

Sustainable Development Goal (SDG) 3, target 3.1, aims to reduce the global maternal mortality ratio to fewer than 70 deaths per 100,000 live births by 2030 [1]. Achieving this goal depends significantly on skilled birth attendant (SBA) delivery [2,3]. SBA delivery, defined as childbirth assisted by a trained healthcare professional such as doctors, midwives, and nurses, is one of the World Health Organization's (WHO) four key pillars of the Safe Motherhood Initiative aimed at reducing maternal mortality [4–7]. By providing a safe and supportive environment during delivery, SBA helps minimize complications for both mothers and newborns. Globally, about two-thirds of births are attended by skilled health personnel [8,9].

Between 2000 and 2023, the global maternal mortality ratio (MMR, maternal deaths per 100,000 live births) declined by about 40%, yet over 90% of maternal deaths occurred in low- and lower-middle-income countries in 2023. Although maternal mortality decreased from 385 in 1990–216 in 2015, and further to 197 per 100,000 live births in 2023, Sub-Saharan Africa alone accounted for approximately 70% of maternal deaths (182,000) [10–13]. In Lesotho, while 87% of births are attended by skilled health personnel, the maternal mortality ratio (MMR) remained high at approximately 566.2 deaths per 100,000 live births in 2020 [14]. According to the World Health Organization (WHO), Lesotho's MMR decreased by nearly 10%, from 566.2 to 478 deaths per 100,000 live births between 2020 and 2023, with an estimated 267 maternal deaths in 2023. Despite this progress, the MMR remains above the Sub-Saharan African average of 442 per 100,000 live births, highlighting the need for continued efforts to achieve the global target of fewer than 70 maternal deaths per 100,000 live births by 2030 [15]. Several national efforts, including the implementation of community-based maternal health programs and initiatives

to improve antenatal care coverage, have been launched to improve maternal health outcomes in Lesotho. However, geographical barriers, limited healthcare infrastructure in rural areas, and socioeconomic disparities continue to hinder equitable access to skilled delivery services [14].

Previous studies have identified multiple factors significantly influencing SBA delivery, including maternal age, education, occupation, marital status, husband's education, media exposure, wealth index, decision-making autonomy, health insurance coverage, maternal age at first birth, time and number of antenatal care (ANC) visits, religion, distance to health facility, region, and residency [7,9,16–25]. However, there is a paucity of research utilizing the most recent Demographic and Health Survey (DHS) data to evaluate these determinants comprehensively within the current Lesotho context.

Although Lesotho reports relatively high SBA coverage, maternal mortality remains persistently elevated, suggesting that coverage has not translated into better outcomes and pointing to equity and quality-of-care gaps. Updating the evidence on modifiable and persistent factors influencing SBA utilization is therefore essential. Using the latest nationally representative data, this study examines the determinants of SBA delivery among reproductive-age women in Lesotho to inform more targeted and effective maternal health interventions.

## Method

### Data source, study setting and study design

This study used secondary data from the 2023–2024 Lesotho Demographic and Health Survey (LSDHS), a nationally representative survey conducted across Lesotho. The survey employed a two-stage stratified sampling method. A cross-sectional study design was applied, and the analysis included 1407 women of reproductive age (15–49 years).

### Study variables

**Dependent variable.** The outcome variable was SBA delivery, defined as live births attended by skilled health personnel, including doctors, nurses, or midwives. This variable was dichotomized as "1" for women who reported receiving skilled birth attendance and "0" for those who did not.

**Independent variables.** The Independent variables included in the study were maternal age, maternal education, maternal occupation, marital status, husband's education, media exposure (women were considered to have media exposure if they had access to at least one information source (radio, television, or newspaper)), wealth index, decision-making autonomy, health insurance coverage, maternal age at first birth, time of ANC visit, number of ANC visits, religion, distance to health facility (Respondent's self-reported perception of whether distance to a health facility posed a major problem in accessing maternal healthcare services), region, residency, economic status and community level media exposure.

### Operational definitions

**Distance to health facility:** Whether the distance to the nearest health facility makes it hard for a woman to get medical care when she is sick. It is coded as 1 = big problem and 2 = not a big problem.

**Community level media exposure:** This was measured by the proportion of women within each cluster who were exposed to at least one form of media television, radio, or newspaper. Based on the distribution, community-level media exposure was categorized as "0" for low and "1" for high exposure, following a similar approach used for other community-level variables.

**Economic status:** This was determined by the proportion of women in a cluster who were classified as "poor" or "poorest" based on the wealth index. The percentage of these women was calculated for each cluster to represent the overall poverty level. Clusters were then categorized as having low or high community poverty based on the national median.

## Data analysis

Data analysis was conducted using STATA version 17, incorporating sample weights to account for the complex sampling design of the Demographic and Health Survey (DHS). Descriptive statistics were presented as percentages to summarize the characteristics of the study population. Given the hierarchical structure of the data, multilevel logistic regression analysis was employed, with random effects included to account for the clustering of women within communities and to capture unobserved between-cluster variability. Results were reported as Adjusted Odds Ratios (AOR) with 95% Confidence Intervals (CI), and a p-value of less than 0.05 was considered statistically significant. Four models were fitted: the Null Model (with no predictors), Model I (including individual-level factors), Model II (including community-level factors), and Model III (including both individual- and community-level factors). The model with the lowest deviance and highest log-likelihood was identified as the best-fitting model.

## Ethical consideration

Access to the data was granted by the DHS Program following an online request submitted through their official platform.

## Result

### Background characteristics of the study population

A total weighted sample of 1407 reproductive-age women in Lesotho were included in this study. The majority were aged 25–34 years (42.61%) and had attained at least secondary education (58.54%). Similarly, among their husbands, 43.79% had attained secondary education. Most women were not employed (70.60%) and were either married or living with a partner (73.10%). 39.96 and 39.25% of the participants belonged to poor and rich households respectively. A large proportion had media exposure (73.88%) and reported having decision-making autonomy (92.10%). However, 98.26% lacked health insurance coverage, and 75.86% did not perceive distance to health facilities as a big problem.

The largest religious group among participants was Roman Catholic (35.02%), and approximately one-third (31.84%) resided in the Maseru region. More than half (61.56%) lived in rural areas. At the community level, the majority lived in areas characterized by low poverty (54.17%) and low media exposure (59.53%) (Table 1).

### Reproductive health characteristics of the study population

The majority of reproductive-age women in Lesotho had their first birth before the age of 19 (46.12%). Additionally, 55.87% initiated ANC during the first trimester, 71.82% had more than four ANC visits and 1,292 women were assisted by SBA during delivery (Table 1).

### Random effect analysis and model comparison of SBA delivery

The intracluster correlation coefficient (ICC) in the null model indicated that 16.5% of the variance in SBA delivery was attributable to differences between clusters. The median odds ratio (MOR) was 2.16, suggesting that women residing in clusters with a higher prevalence of SBA delivery had 2.16 times greater odds of receiving assistance from a SBA compared to those in clusters with lower prevalence. This reflects significant heterogeneity across clusters. Among the models tested, Model III was identified as the best-fitting model, as demonstrated by the lowest deviance value and highest log-likelihood (Table 2).

### Fixed effect analysis of SBA delivery

In the bi-variable multilevel logistic regression analysis, maternal age, maternal education, husband's education, media exposure, wealth index, maternal age at first birth, time of ANC visit, distance to health facility, residency, Economic status, and community-level media exposure were found to be significantly associated with SBA delivery

**Table 1. Sociodemographic characteristics of the study participants of Lesotho, 2023–2024 LSDHS (N = 1407).**

| Variables | Category | Frequency (weighted) | Percent |
|---|---|---|---|
| **Age** | 15-24 | 570 | 40.54 |
| | 25-34 | 599 | 42.61 |
| | 35-49 | 238 | 16.86 |
| **Maternal education** | No education | 10 | 0.69 |
| | Primary | 366 | 26.00 |
| | Secondary | 823 | 58.54 |
| | Higher | 208 | 14.77 |
| **Maternal occupation** | Not working | 993 | 70.60 |
| | Working | 414 | 29.40 |
| **Marital status** | Never married | 238 | 16.87 |
| | Married & living with partner | 1,028 | 73.10 |
| | Widowed, divorce & separated | 141 | 10.03 |
| **Husband's education** | No education | 81 | 7.84 |
| | Primary | 364 | 35.42 |
| | Secondary | 450 | 43.79 |
| | Higher | 133 | 12.95 |
| **Media exposure** | No | 367 | 26.12 |
| | Yes | 1,040 | 73.88 |
| **Wealth index** | Poor | 562 | 39.96 |
| | Middle | 293 | 20.80 |
| | Rich | 552 | 39.25 |
| **Decision-making autonomy** | Yes | 947 | 92.10 |
| | No | 81 | 7.90 |
| **Health insurance coverage** | Yes | 25 | 1.74 |
| | No | 1,382 | 98.26 |
| **SBA delivery** | No | 115 | 8.20 |
| | Yes | 1,292 | 91.80 |
| **Maternal age at 1st birth** | <19 | 642 | 46.12 |
| | 20-24 | 508 | 36.53 |
| | >=25 | 242 | 17.36 |
| **Time of ANC visit** | First trimester | 762 | 55.87 |
| | Second trimester | 514 | 37.74 |
| | Third trimester | 87 | 6.38 |
| **Number of ANC visits** | No visit | 44 | 3.11 |
| | 1–4 visits | 353 | 25.07 |
| | >4 visits | 1,010 | 71.82 |
| **Religion** | Roman catholic | 493 | 35.02 |
| | Lesotho evangelical church | 171 | 12.19 |
| | Methodist | 21 | 1.45 |
| | Anglican church | 81 | 5.73 |
| | Seventh day Adventist | 19 | 1.35 |
| | Pentecostal | 243 | 17.31 |
| | Other Christian | 353 | 25.09 |
| | Other | 26 | 1.86 |
| **Distance to health facility** | Big problem | 378 | 26.86 |
| | Not big problem | 1,029 | 73.14 |

*(Continued)*

 

**Table 1.** (Continued)

| Variables | Category | Frequency (weighted) | Percent |
|---|---|---|---|
| **Region** | Butha-buthe | 90 | 6.42 |
| | Leribe | 234 | 16.60 |
| | Berea | 196 | 13.94 |
| | Maseru | 448 | 31.84 |
| | Mafeteng | 77 | 5.48 |
| | Mohale's hoek | 78 | 5.53 |
| | Quthing | 53 | 3.76 |
| | Qacha's nek | 47 | 3.32 |
| | Mokhotlong | 68 | 4.85 |
| | Thaba-tseka | 116 | 8.25 |
| **Residency** | Urban | 541 | 38.44 |
| | Rural | 866 | 61.56 |
| **Economic status** | Low | 762 | 54.17 |
| | High | 645 | 45.83 |
| **Community level media exposure** | Low | 838 | 59.53 |
| | High | 569 | 40.47 |

**Table 2. Random effects and model comparison of SBA delivery.**

| Parameters | Null model | Model I | Model II | Model III |
|---|---|---|---|---|
| Cluster level Variance | 0.65 | 0.63 | 0.55 | 0.56 |
| ICC | 16.5% | 16.2% | 14.4% | 14.5% |
| PCV | Reference | 3.08% | 15.39% | 13.85% |
| MOR | 2.16 | 2.13 | 2.03 | 2.04 |
| Deviance | 784.85 | 376.56 | 763.12 | 373.19 |
| LLR | −392.42 | −188.28 | −381.56 | −186.59 |

ICC: intracluster correlation coefficient; PCV: proportional change in variance; MOR: median odds ratio; LLR: log-likelihood ratio.

at a p-value of <0.2. These variables were included in the multivariable analysis. In the final model, maternal age, maternal education, wealth index, maternal age at first birth, and time of ANC visit remained significantly associated with SBA delivery (Table 3).

Women aged 35–49 had 60% lower odds of receiving SBA delivery compared to those aged 15–24 (AOR = 0.40; 95% CI: 0.17, 0.95). Women who attained primary education (AOR = 1.47; 95% CI: 1.02, 4.90), secondary education (AOR = 1.63; 95% CI: 1.12, 2.94), and tertiary education (AOR = 3.88; 95% CI: 2.50, 5.20) had 1.47, 1.63, and 3.88 times higher odds, respectively, of receiving SBA delivery compared to women with no formal education.

Women from rich households were 4.30 times more likely to utilize SBA delivery services than those from poor households (AOR = 4.30; 95% CI: 1.10, 16.80). Furthermore, women whose first birth occurred between the ages of 20–24 had 3.74 times higher odds of receiving SBA delivery compared to those who gave birth before age 19 (AOR = 3.74; 95% CI: 1.77, 7.89).

In contrast, women who initiated ANC in the third trimester had 63% lower odds of receiving SBA delivery compared to those who began ANC in the first trimester (AOR = 0.37; 95% CI: 0.14, 0.99).

**Table 3. Multilevel logistic regression analysis of SBA delivery Lesotho, 2023–2024 LSDHS.**

| Variables | Model 1 | Model 2 | Model 3 |
|---|---|---|---|
| **Age** | | | |
| 15-24 | 1 | – | 1 |
| 25-34 | 0.61 (0.30, 1.24) | – | 0.57 (0.28,1.16) |
| 35-49 | 0.38 (0.16, 0.90)* | – | 0.40 (0.17, 0.95)* |
| **Maternal education** | | | |
| No education | 1 | – | 1 |
| Primary | 1.53 (1.08,4.82)* | – | 1.47 (1.02, 4.90)* |
| Secondary | 1.8 (1.11,2.41)* | – | 1.63(1.12, 2.94)* |
| Higher | 3.78 (2.12,4.62)* | – | 3.88 (2.50,5.20)* |
| **Husband's education** | | | |
| No education | 1 | – | 1 |
| Primary | 0.85 (0.35, 2.03) | – | 0.89 (0.37, 2.13) |
| Secondary | 2.34 (0.80, 6.81) | – | 2.52 (0.86, 7.34) |
| Higher | 3.84 (0.16, 95.10) | – | 4.08 (0.17, 10.8) |
| **Media exposure** | | | |
| No | 1 | – | 1 |
| Yes | 1.16 (0.60, 2.24) | – | 1.37(0.68, 2.74) |
| **Wealth index** | | | |
| Poor | 1 | – | 1 |
| Middle | 2.18 (0.92, 5.14) | – | 2.15 (0.85,5.49) |
| Rich | 4.20 (1.39,12.64)* | – | 4.30(1.10, 16.80)* |
| **Maternal age at 1st birth** | | | |
| <19 | 1 | – | 1 |
| 20-24 | 3.58 (1.70,7.54)** | – | 3.74 (1.77, 7.89)** |
| >=25 | 6.22 (1.18, 10.76)* | – | 6.50 (1.22, 10.81) |
| **Time of ANC visit** | | | |
| First trimester | 1 | – | 1 |
| Second trimester | 0.63 (0.34,1.17) | – | 0.63 (0.34,1.17) |
| Third trimester | 0.36 (0.14, 0.91)* | – | 0.37 (0.14, 0.99)* |
| **Distance to health facility** | | | |
| Big problem | – | 1 | 1 |
| Not big problem | – | 0.94 (0.59,1.51) | 0.78 (0.40, 1.52) |
| **Residency** | | | |
| Urban | – | 1 | |
| Rural | – | 0.71 (0.34,1.44) | 1.14 (0.35, 3.74) |
| **Economic status** | | | |
| Low | – | 1 | 1 |
| High | – | 0.43 (0.21,0.87)* | 0.54 (0.19,1.52) |
| **Community-level media exposure** | | | |
| Low | – | 1 | 1 |
| High | – | 0.97 (0.53,1.76) | 1.9 (0.79,4.57) |

Significant at ** = p < 0.001, *=p < 0.05.

## Discussion

Based on data from the 2023–2024 Lesotho Demographic and Health Survey (LSDHS), this study investigated the factors influencing SBA using a multilevel analysis. Maternal age, maternal educational status, household wealth index, age at first birth, and timing of ANC visits were significant determinants of SBA delivery.

The prevalence of SBA delivery was 91.8%, which is comparable to findings from a study conducted in 12 East African countries, where the pooled prevalence in Rwanda and Malawi was 90.68% and 89.8%, respectively [9]. However, this figure is higher than those reported in other previous studies done in Ethiopia (49.6 and 50.9%) and Nepal (48%) [16,26,27]. The variation in SBA prevalence across studies may be attributed to differences in study populations, healthcare infrastructure, accessibility and quality of maternal health services, socio-cultural practices, and national policies promoting facility-based deliveries. Lesotho's relatively higher coverage might also reflect the country's progress in implementing maternal health programs and promoting institutional delivery services over the past decade.

In this study, older women (AOR = 0.40) were less likely to receive SBA compared to those aged 15–24. This finding aligns with studies conducted in Ghana [28], Kenya [29], Nepal (2005), and other research carried out in developing countries between 1990 and 2000 [30,31], but contrasts with findings from studies in East African countries [9] and Nepal (2011) [32]. The observed disparity may be attributed to differences in study settings, sample sizes, periods of data collection, and access to services. Moreover, the lower utilization of SBA among older women may result from previous negative experiences with health facilities or a sense of self-reliance developed through prior childbirths [33–35]. Nevertheless, despite their experience, older women remain at increased risk of maternal mortality, highlighting the critical importance of skilled care during childbirth.

In line with previous studies done in Ethiopia, Ghana, Bangladesh, Vietnam, Togo, Uganda, and sub-Saharan Africa countries [9,17,19,20,36–43], women with a primary or higher level of education (AOR = 1.47, 1.63 and 3.88) were more likely to utilize SBA services. This association can be attributed to several factors. Educated women are more likely to have increased health literacy, enabling them to understand the importance of maternal healthcare and recognize danger signs during pregnancy and childbirth. They are also more likely to be exposed to health information through various media, engage in informed decision-making, and navigate the healthcare system effectively. Furthermore, education often empowers women by increasing their autonomy, confidence, and communication skills, which can improve their ability to negotiate and advocate for healthcare needs within the household and community. These factors collectively enhance the likelihood of seeking timely and appropriate maternal health services, including delivery assisted by skilled personnel.

Women from wealthier households (AOR = 4.30) had a higher likelihood of utilizing SBA services, which is consistent with findings from studies conducted in Bangladesh [36,44,45], Vietnam [41], India [46], Nigeria [47], Kenya [18], Togo [42], Ethiopia [16,17,48–51], Ghana [19,52,53] and East Africa countries [9]. This association may be explained by the fact that women in higher wealth quintiles are better able to cover indirect costs related to delivery, such as transportation, medications, and other out-of-pocket expenses. They are also more likely to have access to health-related information, including the advantages of SBA. Furthermore, economically empowered women often have greater autonomy in reproductive health decision-making, which enhances their utilization of skilled delivery services [28,54,55]. In contrast, women from the poor households face significant financial constraints that limit their access to essential maternal healthcare, reducing their likelihood of receiving SBA services.

Women whose first birth occurred between the ages of 20–24 (AOR = 3.74) had higher odds of receiving SBA, which is consistent with findings from 29 sub-Saharan African countries and Pakistan [56,57]. However, this contrasts with studies from Ethiopia, Nigeria and Mali that found no statistically significant association between age at first birth and SBA utilization [16,58,59]. This disparity may be attributed to differences in study settings, sample sizes, socio-cultural norms, and variations in access to maternal health services. Additionally, women who give birth at a younger age may lack the knowledge, confidence, or autonomy needed to seek skilled care. In contrast, women who give birth at older ages are more likely to be informed, empowered, and better equipped to make decisions regarding their maternal health.

Women who began ANC in the third trimester (AOR = 0.37) were less likely to utilize SBA, consistent with findings from Ethiopian studies indicating that initiating ANC in the first or second trimester significantly increases the likelihood of SBA delivery [7,60]. This may be because early ANC visits provide more opportunities for health education, birth preparedness, and awareness of the importance of delivering with skilled assistance. In contrast, late initiation limits these opportunities and may reflect underlying barriers such as limited awareness, poor access to services, or low perceived need for care.

This study provides important insights into the factors influencing SBA delivery in Lesotho. It utilized a large, population-based dataset, ensuring both representativeness and the reliability of national estimates. Furthermore, the application of multilevel modeling strengthens the analysis by accounting for hierarchical data structure. However, the cross-sectional design limits the ability to establish causal relationships. In addition, the use of self-reported data may introduce recall and social desirability biases, particularly given the community-based nature of the survey. The absence of time-sensitive variables, such as recent policy changes or shifts in individual experiences, also narrows the scope of the findings.

## Conclusion

While Lesotho has achieved high levels of skilled birth attendance, important inequalities persist. Maternal age, educational attainment, household wealth, age at first birth, and the timing of ANC initiation remain significant predictors of SBA utilization. Although the country has made notable progress in reducing disparities between advantaged and disadvantaged groups, gaps continue to affect specific populations—particularly women who were very young at the time of their first birth, older mothers, and those who begin ANC late. Eliminating these remaining inequities will require continued efforts to ensure that all women can access timely and appropriate maternal health services.

Focused, multisectoral strategies are essential. Enhancing women's educational and economic opportunities, strengthening community-level support mechanisms, and improving both the accessibility and quality of maternity care can help promote more equitable SBA use. Prioritizing interventions that reach the poorest households and women who had their first birth at a young age will be critical for meeting national and international maternal health goals.

Furthermore, there is a need for more in-depth qualitative research to better understand the barriers and facilitators of SBA utilization in the Lesotho context. Such studies could examine sociocultural norms, perceptions of maternal health services, household decision-making processes, and structural or health system factors that influence women's choices. Insights from this research would complement quantitative findings by identifying context-specific enablers and obstacles, informing targeted interventions, and guiding policies aimed at reducing inequities in maternal health service use. By capturing the lived experiences and perspectives of women, their families, and healthcare providers, qualitative research can generate nuanced evidence to strengthen strategies for improving equitable access to skilled birth attendance.

## Supporting information

**S1 File. Lesotho 2023−24 DHS data.**
(XLS)

## Acknowledgments

We would like to thank the measure DHS program for their permission to access the 2023−24 LSDHS datasets.

## Author contributions

**Conceptualization:** Hiwot Tezera Endale, Tseganesh Asefa, Tiget Ayelgn Mengstie, Mihret Getnet, Helen Lamesgin Endalew, Amare Belete Getahun, Gashaw Dessie, Thomas kidanemariam Yewodiaw, Engidaw Fentahun Enyew.

**Data curation:** Hiwot Tezera Endale, Tseganesh Asefa, Tiget Ayelgn Mengstie, Mihret Getnet, Helen Lamesgin Endalew, Thomas kidanemariam Yewodiaw, Engidaw Fentahun Enyew.

**Formal analysis:** Hiwot Tezera Endale, Tseganesh Asefa, Tiget Ayelgn Mengstie, Mihret Getnet, Amare Belete Getahun, Engidaw Fentahun Enyew.

**Funding acquisition:** Hiwot Tezera Endale, Engidaw Fentahun Enyew.

**Investigation:** Hiwot Tezera Endale, Engidaw Fentahun Enyew.

**Methodology:** Hiwot Tezera Endale, Tseganesh Asefa, Tiget Ayelgn Mengstie, Mihret Getnet, Helen Lamesgin Endalew, Amare Belete Getahun, Gashaw Dessie, Thomas kidanemariam Yewodiaw, Engidaw Fentahun Enyew.

**Project administration:** Hiwot Tezera Endale, Engidaw Fentahun Enyew.

**Resources:** Hiwot Tezera Endale, Engidaw Fentahun Enyew.

**Software:** Hiwot Tezera Endale, Tseganesh Asefa, Tiget Ayelgn Mengstie, Mihret Getnet, Helen Lamesgin Endalew, Amare Belete Getahun, Gashaw Dessie, Thomas kidanemariam Yewodiaw, Engidaw Fentahun Enyew.

**Supervision:** Hiwot Tezera Endale, Tseganesh Asefa, Mihret Getnet, Helen Lamesgin Endalew, Amare Belete Getahun, Gashaw Dessie, Engidaw Fentahun Enyew.

**Validation:** Hiwot Tezera Endale, Tseganesh Asefa, Amare Belete Getahun, Gashaw Dessie, Thomas kidanemariam Yewodiaw, Engidaw Fentahun Enyew.

**Visualization:** Hiwot Tezera Endale, Engidaw Fentahun Enyew.

**Writing – original draft:** Hiwot Tezera Endale.

**Writing – review & editing:** Hiwot Tezera Endale, Tseganesh Asefa, Tiget Ayelgn Mengstie, Mihret Getnet, Helen Lamesgin Endalew, Amare Belete Getahun, Gashaw Dessie, Thomas kidanemariam Yewodiaw, Engidaw Fentahun Enyew.

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
