## [Decision Letter · Decision Letter 0]

19 Sep 2025

Dear Dr. Endale,

Thank you for submitting your manuscript to PLOS ONE. After careful consideration, we feel that it has merit but does not fully meet PLOS ONE’s publication criteria as it currently stands. Therefore, we invite you to submit a revised version of the manuscript that addresses the points raised during the review process.

We look forward to receiving your revised manuscript.

Kind regards,

Sabita Tuladhar, PhD, MHealSc, MA

Academic Editor

PLOS ONE

Journal Requirements:

2. We note that your Data Availability Statement is currently as follows: [All relevant data are within the manuscript and its Supporting Information files]

3. Please include captions for your Supporting Information files at the end of your manuscript, and update any in-text citations to match accordingly. Please see our Supporting Information guidelines for more information: http://journals.plos.org/plosone/s/supporting-information .

4. We are unable to open your Supporting Information file [SBA.DTA]. Please kindly revise as necessary and re-upload.

Additional Editor Comments:

Reviewer #1:

I want to thank the authors for their valuable contribution and the extensive use of a large dataset in their work. However, the manuscript would benefit from substantial revisions. Specifically, the Introduction, Results, Discussion, and Conclusion sections need to be improved. Additionally, the paper contains multiple editing errors that should be addressed to enhance overall readability and scientific rigor. I have also listed some of the comments as shown below.

Title: The title seems like for the case-control, I recommend a little bit to modify

Abstract: List the abbreviation of SBA when you use in the first line of the sentences (Line 38 & 233) and similar comment throughout the paper. In this section mention the AOR for the factors positively associated with SBA. In the recommendation what are the targeted polices (it is general, but the other recommended ideas are good).

Introduction: Although the authors highlighted several important points in the introduction, they failed to adequately address the issue of maternal death and the SBA, particularly by omitting recent evidence and trends from global to regional (Africa) levels. Also cite the reference for Line 63-64 (for the maternal death in Lesotho).

Methods: Change “methods and Materials” to Method; In the operational definition I recommend changing “Community Poverty Level” to Economic status or other word; the media exposure is also not clear for readers. Change “Data management and analysis” to Data analysis. Change “Ethics approval and consent to participants” to Ethical consideration

Result section: I recommend categorizing into background and reproductive health related characteristics separately with subhead instead of “descriptive characteristics”

Discussion: In the discussion of the magnitude of skilled birth attendant (SBA) utilization, comparisons with both higher and lower prevalence rates reported in other studies should be presented and briefly discussed. These comparisons should include logical justifications for the observed differences relative to the current findings. For instance, in Line 175, only a lower prevalence was mentioned, and the cited reference lacks clarity, making it difficult for readers to interpret the comparison effectively.

The discussion in the factors section is well-articulated; however, there are a few minor points that require attention. Specifically, I recommend the authors include the AOR for each factor discussed to provide clear information to readers. Additionally, the study location should be mentioned for each referenced study. For example, in Line 180, references 28 and 29, the geographical area where each study was conducted should be specified. I recommend Similar revisions for Line 210, Line 212, and other sections where contributing factors are discussed.

Conclusion: “Addressing these challenges through targeted policies and programs is essential. Strategies aimed at improving healthcare accessibility, availability, and financial support should be prioritized to enhance SBA coverage and ultimately improve maternal health outcomes.”

The statement is general; it would be more effective to recommend specific actions based on the findings.

Table 1: The number of participants is 1407 but for the variables Age, marital status, Meidia exposure, health insurance coverage, SBA, community level media exposure is 1406.

Distance to health facility: what do you mean ‘Big problem or No big problem”

What is the difference between media exposure and community level media exposure in this paper?

Reviewer #2:

Thank you for this excellent paper. It has analysed recent data resulting in succinct and readable paper with appropriate conclusions. You have speculated on some of the possible reasons behind the findings. This is important because it is only the answers to those questions of "why did a certain group act in a certain way" that can provide real understanding and lead to necessary interventions. I would have suggested that in the conclusion you might have alluded to a need for more detailed qualitative research to be conducted to further elucidate this point, but this is not a criticism. I rarely find that I can make a recommendation for a paper to be accepted without alteration but in this case I feel this recommendation to accept is well justified.

Reviewers' comments:

Reviewer's Responses to Questions

**Comments to the Author**

1. Is the manuscript technically sound, and do the data support the conclusions?

Reviewer #1: Yes

Reviewer #2: Yes

2. Has the statistical analysis been performed appropriately and rigorously?

Reviewer #1: Yes

Reviewer #2: Yes

3. Have the authors made all data underlying the findings in their manuscript fully available?

Reviewer #1: Yes

Reviewer #2: Yes

4. Is the manuscript presented in an intelligible fashion and written in standard English?

Reviewer #1: Yes

Reviewer #2: Yes

Reviewer #1: I want to thank the authors for their valuable contribution and the extensive use of a large dataset in their work. However, the manuscript would benefit from substantial revisions. Specifically, the Introduction, Results, Discussion, and Conclusion sections need to be improved. Additionally, the paper contains multiple editing errors that should be addressed to enhance overall readability and scientific rigor. I have also listed some of the comments as shown below.

Title: The title seems like for the case-control, I recommend a little bit to modify

Abstract: List the abbreviation of SBA when you use in the first line of the sentences (Line 38 & 233) and similar comment throughout the paper. In this section mention the AOR for the factors positively associated with SBA. In the recommendation what are the targeted polices (it is general, but the other recommended ideas are good).

Introduction: Although the authors highlighted several important points in the introduction, they failed to adequately address the issue of maternal death and the SBA, particularly by omitting recent evidence and trends from global to regional (Africa) levels. Also cite the reference for Line 63-64 (for the maternal death in Lesotho).

Methods: Change “methods and Materials” to Method; In the operational definition I recommend changing “Community Poverty Level” to Economic status or other word; the media exposure is also not clear for readers. Change “Data management and analysis” to Data analysis. Change “Ethics approval and consent to participants” to Ethical consideration

Result section: I recommend categorizing into background and reproductive health related characteristics separately with subhead instead of “descriptive characteristics”

Discussion: In the discussion of the magnitude of skilled birth attendant (SBA) utilization, comparisons with both higher and lower prevalence rates reported in other studies should be presented and briefly discussed. These comparisons should include logical justifications for the observed differences relative to the current findings. For instance, in Line 175, only a lower prevalence was mentioned, and the cited reference lacks clarity, making it difficult for readers to interpret the comparison effectively.

The discussion in the factors section is well-articulated; however, there are a few minor points that require attention. Specifically, I recommend the authors include the AOR for each factor discussed to provide clear information to readers. Additionally, the study location should be mentioned for each referenced study. For example, in Line 180, references 28 and 29, the geographical area where each study was conducted should be specified. I recommend Similar revisions for Line 210, Line 212, and other sections where contributing factors are discussed.

Conclusion: “Addressing these challenges through targeted policies and programs is essential. Strategies aimed at improving healthcare accessibility, availability, and financial support should be prioritized to enhance SBA coverage and ultimately improve maternal health outcomes.”

The statement is general; it would be more effective to recommend specific actions based on the findings.

Table 1: The number of participants is 1407 but for the variables Age, marital status, Meidia exposure, health insurance coverage, SBA, community level media exposure is 1406.

Distance to health facility: what do you mean ‘Big problem or No big problem”

What is the difference between media exposure and community level media exposure in this paper?

Reviewer #2: Thank you for this excellent paper. It has analysed recent data resulting in succinct and readable paper with appropriate conclusions. You have speculated on some of the possible reasons behind the findings. This is important because it is only the answers to those questions of "why did a certain group act in a certain way" that can provide real understanding and lead to necessary interventions. I would have suggested that in the conclusion you might have alluded to a need for more detailed qualitative research to be conducted to further elucidate this point, but this is not a criticism. I rarely find that I can make a recommendation for a paper to be accepted without alteration but in this case I feel this recommendation to accept is well justified.

**Do you want your identity to be public for this peer review?** For information about this choice, including consent withdrawal, please see our Privacy Policy

Reviewer #1: No

Reviewer #2: No

---

## [Author Response · Author response to Decision Letter 1]

8 Oct 2025

Date: October 07, 2025

PLOS ONE

Manuscript tittle: Determinants of skilled birth attendant delivery among reproductive age women in Lesotho: a multilevel analysis.

Submission ID: PONE-D-25-34001

Response to Journal Requirements:

Authors’ response: We have ensured that our manuscript aligns with PLOS ONE's style requirements, following the templates provided. The revised version adheres to the required formatting and file-naming conventions.

2. We note that your Data Availability Statement is currently as follows: [All relevant data are within the manuscript and its Supporting Information files]. Please confirm at this time whether or not your submission contains all raw data required to replicate the results of your study.

Authors’ response: We confirm that all relevant data required to replicate the study findings are included in the manuscript and supporting information files. We will provide them or deposit the dataset in a public repository as recommended.

3. Please include captions for your Supporting Information files at the end of your manuscript, and update any in-text citations to match accordingly.

Authors’ response: Thank you for this suggestion. We have added caption for the Supporting Information file at the end of the manuscript.

4. We are unable to open your Supporting Information file [SBA.DTA]. Please kindly revise as necessary and re-upload.

Authors’ response: Thank you for bringing this to our attention. The Supporting Information file [SBA.DTA] is a Stata dataset file, which requires Stata software to open. We have re-checked the file and confirmed that it opens correctly in Stata. The file has been re-uploaded and renamed as Lesotho 2023-24 DHS data.DTA to ensure it can be accessed without any issues.

Authors’ response: Thank you for bringing this to our attention. We note that no specific publications were suggested for citation, and therefore no additional references were added.

Point by point response to Reviewers’ comments

Reviewer #1:

Comment 1: The title seems like for the case-control, I recommend a little bit to modify

Authors’ response: Thank you for your valuable comment. We have revised the title to accurately reflect the study design: as “Determinants of skilled birth attendant delivery among reproductive age women in Lesotho: a multilevel analysis of demographic and health survey data”

Comment 2: List the abbreviation of SBA when you use in the first line of the sentences (Line 38 & 233) and similar comment throughout the paper. In this section mention the AOR for the factors positively associated with SBA. In the recommendation what are the targeted polices (it is general, but the other recommended ideas are good).

Authors’ response: Thank you for your insightful comment. We have defined “SBA” at first use and throughout the manuscript. AORs for statistically significant predictors have been added in the abstract. The recommendations were revised to include specific actions.

Comment 3: Although the authors highlighted several important points in the introduction, they failed to adequately address the issue of maternal death and the SBA, particularly by omitting recent evidence and trends from global to regional (Africa) levels. Also cite the reference for Line 63-64 (for the maternal death in Lesotho).

Authors’ response: Thank you for your comment. We expanded the Introduction to include updated statistics on maternal mortality and SBA trends globally, in sub-Saharan Africa, and in Lesotho. We also cite the references for line 63–64.

Comment 4: Change “methods and Materials” to Method; In the operational definition I recommend changing “Community Poverty Level” to Economic status or other word; the media exposure is also not clear for readers. Change “Data management and analysis” to Data analysis. Change “Ethics approval and consent to participants” to Ethical consideration

Authors’ response: Thank you very much for your constructive comments. We tried to incorporate the comment in the manuscript.

Comment 5: I recommend categorizing into background and reproductive health related characteristics separately with subhead instead of “descriptive characteristics”.

Authors’ response: Thank you for your comment. We tried to incorporate the comment in the manuscript.

Comment 6: In the discussion of the magnitude of skilled birth attendant (SBA) utilization, comparisons with both higher and lower prevalence rates reported in other studies should be presented and briefly discussed. These comparisons should include logical justifications for the observed differences relative to the current findings. For instance, in Line 175, only a lower prevalence was mentioned, and the cited reference lacks clarity, making it difficult for readers to interpret the comparison effectively. The discussion in the factors section is well-articulated; however, there are a few minor points that require attention. Specifically, I recommend the authors include the AOR for each factor discussed to provide clear information to readers. Additionally, the study location should be mentioned for each referenced study. For example, in Line 180, references 28 and 29, the geographical area where each study was conducted should be specified. I recommend Similar revisions for Line 210, Line 212, and other sections where contributing factors are discussed

Authors’ response: Thank you for this valuable comment. We have revised the discussion to compare our findings with available published studies and included possible explanations for the observed differences. Since there were no studies reporting higher SBA prevalence in comparable contexts, we compared our findings with those reporting similar or lower prevalence rates. Adjusted odds ratios (AORs) for each discussed factor were also incorporated to enhance clarity. Each cited reference now clearly specifies the country or region of origin to improve clarity and context.

Comment 7: “Addressing these challenges through targeted policies and programs is essential. Strategies aimed at improving healthcare accessibility, availability, and financial support should be prioritized to enhance SBA coverage and ultimately improve maternal health outcomes.”

The statement is general; it would be more effective to recommend specific actions based on the findings.

Authors’ response: Thank you for your valuable comment. We revised the Conclusion to provide actionable recommendations based on our findings.

Comment 8: Table 1: The number of participants is 1407 but for the variables Age, marital status, Media exposure, health insurance coverage, SBA, community level media exposure is 1406. Distance to health facility: what do you mean ‘Big problem or No big problem”

What is the difference between media exposure and community level media exposure in this paper?

Authors’ response: Thank you for your comment. The minor inconsistency in sample size occurred because we used weighted frequencies, which produced decimals; these have now been rounded to the nearest whole number. The variable Distance to health facility has been clarified to reflect the respondent’s self-reported perception of whether distance to a health facility posed a major problem in accessing maternal healthcare services. Additionally, the operational definitions of media exposure (individual-level exposure to radio, television, or newspapers) and community-level media exposure (the aggregated proportion of women in the community reporting media exposure) have been clearly specified in the Methods section.

Reviewer #2:

Reviewers’ comment: Thank you for this excellent paper. It has analysed recent data resulting in succinct and readable paper with appropriate conclusions. You have speculated on some of the possible reasons behind the findings. This is important because it is only the answers to those questions of "why did a certain group act in a certain way" that can provide real understanding and lead to necessary interventions. I would have suggested that in the conclusion you might have alluded to a need for more detailed qualitative research to be conducted to further elucidate this point, but this is not a criticism. I rarely find that I can make a recommendation for a paper to be accepted without alteration but in this case I feel this recommendation to accept is well justified.

Authors’ response: We sincerely thank the reviewer for the thoughtful and positive feedback. We greatly appreciate the recognition of the clarity, readability, and relevance of our analysis. We also value your suggestion regarding the potential for further qualitative research to explore the underlying reasons behind our findings. To address this, we have included a sentence in the Conclusion highlighting the need for more detailed qualitative studies to better understand the factors influencing skilled birth attendant utilization. We are grateful for your recommendation to accept our manuscript.

---

## [Decision Letter · Decision Letter 1]

19 Nov 2025

PONE-D-25-34001R1

Determinants of skilled birth attendant delivery among reproductive age women in Lesotho: a multilevel analysis of Demographic and Health Survey Data

PLOS ONE

Dear Dr. Endale,

Thank you for submitting your manuscript to PLOS ONE. After careful consideration, we feel that it has merit but does not fully meet PLOS ONE’s publication criteria as it currently stands. Therefore, we invite you to submit a revised version of the manuscript that addresses the points raised during the review process.

We have received a few additional comments on your manuscript. We therefore invite you to submit a revised version that addresses the points raised during the review process.

We look forward to receiving your revised manuscript.

Kind regards,

Sabita Tuladhar, PhD, MHealSc, MA

Academic Editor

PLOS ONE

Journal Requirements:

Reviewers' comments:

Reviewer's Responses to Questions

**Comments to the Author**

Reviewer #3: (No Response)

Reviewer #4: All comments have been addressed

2. Is the manuscript technically sound, and do the data support the conclusions?

Reviewer #3: Yes

Reviewer #4: Yes

3. Has the statistical analysis been performed appropriately and rigorously?

Reviewer #3: I Don't Know

Reviewer #4: I Don't Know

4. Have the authors made all data underlying the findings in their manuscript fully available?

Reviewer #3: Yes

Reviewer #4: Yes

5. Is the manuscript presented in an intelligible fashion and written in standard English?

Reviewer #3: Yes

Reviewer #4: Yes

Reviewer #3: Overall Assessment

The manuscript is well-organized and follows a clear structure; however, several methodological and interpretive issues require further refinement to meet the standards for publication in PLOS ONE. While the topic is relevant and timely, much of the content repeats already well-established findings, and the discussion lacks contextual depth for Lesotho. The clarity, methodological description, and analytical interpretation should be strengthened.

Major Comments

Novelty and Context:

The study does not sufficiently highlight what is new compared to previous evidence. Authors should clearly identify how this analysis adds to the existing body of knowledge on SBA utilization in Lesotho.

Introduction:

The global and country-level maternal mortality figures are inconsistent and somewhat confusing (e.g., 566.2 vs. 529 per 100,000).

Clarify whether these are WHO estimates and ensure consistency between the abstract and introduction.

Define “healthcare professionals” and ensure comparability with the operational definition of SBA.

The rationale should be reframed to justify the study in light of persistently high MMR despite high SBA coverage.

Methods:

Clearly describe why random effect analysis was used and mention it explicitly in the methodology.

Explain the rationale for including specific variables (e.g., community-level indicators).

Ensure consistency in terminology (e.g., “Economic status” vs. “Community poverty level”).

Results:

Present rural–urban differences, as this was not visible in the results despite being an important determinant.

Ensure figures and tables have consistent sample sizes (1407 vs. 1406).

Clarify variable meanings (e.g., “distance to health facility: big problem/no big problem”).

Discussion:

Focus on key findings with policy or programmatic relevance to Lesotho rather than reiterating general determinants.

Some comparative interpretations are contradictory (e.g., Nepal results cited with conflicting conclusions).

The discussion overemphasizes descriptive comparison and lacks deeper explanation of why these associations persist.

Conclusion:

Recommendations are too general. Since SBA coverage is already high, emphasize addressing equity and quality gaps.

Consider aligning conclusions with findings, avoiding overly broad or repetitive policy statements.

Inclusion of a recommendation for future qualitative research is appropriate and should be expanded slightly.

Minor Comments

Ensure consistent abbreviation use.

Improve wording clarity (e.g., “area,” “domain,” “healthcare professionals”).

Check grammar, punctuation, and spacing (comma vs. hyphen use for example in CI in abstract).

Ensure data access and ethics sections meet PLOS ONE requirements precisely.

Reviewer #4: The authors have adequately addressed the comments provided by the previous reviewers. However, the use of abbreviations can be improved, starting from page 1.

**Do you want your identity to be public for this peer review?** For information about this choice, including consent withdrawal, please see our Privacy Policy

Reviewer #3: **Yes: ** Khim Bahadur Khadka

Reviewer #4: **Yes: ** Sabita Tuladhar

---

## [Author Response · Author response to Decision Letter 2]

11 Dec 2025

Point by point response to Reviewers’ comments

Reviewer #3:

Comment 1: The study does not sufficiently highlight what is new compared to previous evidence. Authors should clearly identify how this analysis adds to the existing body of knowledge on SBA utilization in Lesotho.

Authors’ response: Thank you for this valuable comment. We have now strengthened the Introduction sections to more clearly articulate the novelty of our study.

Comment 2: The global and country-level maternal mortality figures are inconsistent and somewhat confusing (e.g., 566.2 vs. 529 per 100,000). Clarify whether these are WHO estimates and ensure consistency between the abstract and introduction. Define “healthcare professionals” and ensure comparability with the operational definition of SBA. The rationale should be reframed to justify the study in light of persistently high MMR despite high SBA coverage.

Authors’ response: Thank you for this comment. We have corrected and aligned the maternal mortality figures in both the abstract and introduction using the latest WHO estimates. We also clarified the definition of “healthcare professionals” to ensure consistency with the WHO-defined operational definition of skilled birth attendants. Additionally, we revised the rationale to emphasize the persistent gap between high SBA coverage and the still elevated maternal mortality ratio in Lesotho, highlighting the need for updated evidence from the 2023–2024 LSDHS.

Comment 3: Clearly describe why random effect analysis was used and mention it explicitly in the methodology. Explain the rationale for including specific variables (e.g., community-level indicators). Ensure consistency in terminology (e.g., “Economic status” vs. “Community poverty level”).

Authors’ response: Thank you for these helpful suggestions. We have now explicitly stated in the methodology that a random-effects (multilevel) analysis was used to account for the hierarchical structure of the DHS data, where individuals are nested within clusters, and to capture unobserved community-level variation in SBA utilization. We also clarified the rationale for including specific community-level variables, noting that they provide important contextual information that may influence access, behaviours, and health service utilization beyond individual characteristics. In addition, we reviewed the manuscript to ensure consistent use of terminology, particularly standardizing the term “economic status.”

Comment 4: Present rural–urban differences, as this was not visible in the results despite being an important determinant. Ensure figures and tables have consistent sample sizes (1407 vs. 1406). Clarify variable meanings (e.g., “distance to health facility: big problem/no big problem”).

Authors’ response: Thank you for these comments. Upon review, we found that rural–urban residence did not show a meaningful difference in SBA utilization and was not significantly associated with the outcome. We also rechecked the tables and confirmed that there are no discrepancies in the reported sample size. Additionally, the meaning of variables such as “distance to health facility” is now clearly defined in the operational definitions section.

Comment 5: Focus on key findings with policy or programmatic relevance to Lesotho rather than reiterating general determinants. Some comparative interpretations are contradictory (e.g., Nepal results cited with conflicting conclusions). The discussion overemphasizes descriptive comparison and lacks deeper explanation of why these associations persist.

Authors’ response: Thank you for these constructive comments. We have focused our discussion on key findings with policy and programmatic relevance to Lesotho and have provided interpretations based on the available data, while acknowledging the limitations inherent in secondary data analysis

Comment 6: Recommendations are too general. Since SBA coverage is already high, emphasize addressing equity and quality gaps. Consider aligning conclusions with findings, avoiding overly broad or repetitive policy statements. Inclusion of a recommendation for future qualitative research is appropriate and should be expanded slightly.

Authors’ response: Thank you for this insightful comment. We have revised the recommendations to focus on addressing equity and quality gaps in SBA utilization, given that coverage is already high in Lesotho. Conclusions have been aligned more closely with the study findings, avoiding overly broad or repetitive policy statements. Additionally, we have expanded the recommendation for future qualitative research to better understand the underlying barriers and facilitators of SBA utilization, including sociocultural, household, and health system factors.

Comment 7: Ensure consistent abbreviation use. Improve wording clarity (e.g., “area,” “domain,” “healthcare professionals”). Check grammar, punctuation, and spacing (comma vs. hyphen use for example in CI in abstract). Ensure data access and ethics sections meet PLOS ONE requirements precisely.

Authors’ response: Thank you for this comment. We have carefully reviewed the manuscript to ensure consistent use of abbreviations throughout. Wording has been clarified for terms such as “area,” “domain,” and “healthcare professionals,” and grammar, punctuation, and spacing including the presentation of confidence intervals in the abstract have been corrected. We also ensured that the data access and ethics statements fully comply with PLOS ONE requirements.

Reviewer #4:

Reviewers’ comment: The authors have adequately addressed the comments provided by the previous reviewers. However, the use of abbreviations can be improved, starting from page 1.

Authors’ response: We sincerely thank the reviewer for the thoughtful and positive feedback. We have carefully reviewed the manuscript and improved the use of abbreviations throughout

---

## [Editor Report · Decision Letter 2]

15 Dec 2025

Determinants of skilled birth attendant delivery among reproductive age women in Lesotho: a multilevel analysis of demographic and health survey data

PONE-D-25-34001R2

Dear Dr. Endale,

We’re pleased to inform you that your manuscript has been judged scientifically suitable for publication and will be formally accepted for publication once it meets all outstanding technical requirements.

Kind regards,

Sabita Tuladhar, PhD, MHealSc, MA

Academic Editor

PLOS One

---

## [Editor Report · Acceptance letter]

PONE-D-25-34001R2

PLOS One

Dear Dr. Endale,

I'm pleased to inform you that your manuscript has been deemed suitable for publication in PLOS One. Congratulations! Your manuscript is now being handed over to our production team.

Kind regards,

on behalf of

Dr. Sabita Tuladhar

Academic Editor

PLOS One